# RNA Flow Cytometry for the Study of T Cell Metabolism

**DOI:** 10.3390/ijms22083906

**Published:** 2021-04-09

**Authors:** Alessandra Rossi, Ilenia Pacella, Silvia Piconese

**Affiliations:** 1Department of Internal Clinical Sciences, Anaesthesiology and Cardiovascular Sciences, Sapienza University of Rome, 00161 Roma, Italy; alessandra.rossi@uniroma1.it (A.R.); ilenia.pacella@uniroma1.it (I.P.); 2Istituto Pasteur Italia-Fondazione Cenci Bolognetti, 00161 Roma, Italy

**Keywords:** T cells, metabolism, mitochondria, RNA, flow cytometry

## Abstract

T cells undergo activation and differentiation programs along a continuum of states that can be tracked through flow cytometry using a combination of surface and intracellular markers. Such dynamic behavior is the result of transcriptional and post-transcriptional events, initiated and sustained by the activation of specific transcription factors and by epigenetic remodeling. These signaling pathways are tightly integrated with metabolic routes in a bidirectional manner: on the one hand, T cell receptors and costimulatory molecules activate metabolic reprogramming; on the other hand, metabolites modify T cell transcriptional programs and functions. Flow cytometry represents an invaluable tool to analyze the integration of phenotypical, functional, metabolic and transcriptional features, at the single cell level in heterogeneous T cell populations, and from complex microenvironments, with potential clinical application in monitoring the efficacy of cancer immunotherapy. Here, we review the most recent advances in flow cytometry-based analysis of gene expression, in combination with indicators of mitochondrial activity, with the aim of revealing and characterizing major metabolic pathways in T cells.

## 1. Introduction

The field of immunometabolism, which is the study of how metabolic pathways affect immune cell functions, is continuously growing [1]. While the analysis of immune cell phenotype, functions and gene expression profile is possible, even at the single cell level, thanks to state-of the-art technologies such as multiparameter flow cytometry and next-generation sequencing, the study of metabolite content and flux in immune cells is more difficult to perform, due to technical constraints such as the amount of required material. However, especially in T cells, much information is available regarding the transcriptional control of metabolism: indeed, key transcription factors promote the expression of genes involved in specific metabolic pathways, thus dictating the immune as well the metabolic program of T cells in different states [2]. Therefore, the analysis of the expression of selected genes may provide relevant information on the metabolic profile of a certain cell type or stage. This analysis can be performed through RNA flow cytometry: this approach, which can be easily performed in any laboratory provided with a flow cytometer, allows for the analysis of selected mRNAs at the single cell level, in combination with other flow cytometry-based assays, such as surface and intracellular marker staining or mitochondrial mass/polarization analysis. In this review, we will first provide an overview of recent knowledge in the field of T cell metabolism, with a special focus on its transcriptional control. Then, we will illustrate how RNA flow cytometry works and how it can be combined with conventional flow cytometry, in order to obtain a complete picture of the immunometabolic profile of T cells at the single-cell level. Finally, we will propose an example of this procedure applied to the study of regulatory T cells (Tregs) and demonstrate the compatibility of RNA flow cytometry with the analysis of intranuclear marker expression and of mitochondrial mass.

## 2. Overview of T Cell Metabolism

Immune cells harbor a high level of plasticity to sustain the broad range of immune functions. A growing body of literature has described a central role for cellular metabolism in regulating the transcriptional reprogramming accompanying the different phases of leukocyte functional states [2]. In T cells, transition through the different activation states is accompanied by the active reprogramming of cellular metabolism. Mature naive T cells rely mainly on oxidative phosphorylation (OXPHOS) to maintain their quiescent state while activated T cells sustain their growth by switching to a glycolytic-lipogenic metabolism, characterized by the increased uptake of glucose (and other nutrients) from the extracellular environment, the generation of ATP from substrate-level phosphorylation during glycolysis, the conversion of pyruvate into lactate, and biosynthesis of macromolecules for cell growth and division. Memory T cells resume the OXPHOS but, as compared to naive cells, they are “metabolically primed” for a glycolytic switch, a combination that ensures both long-term survival and a prompt response to an antigen recall [3,4]. These metabolic changes are directly orchestrated by the triggering of the T cell receptor (TCR) and costimulatory molecules. In naive T cells, the concerted engagement of TCR and CD28 by cognate molecules on antigen presenting cells (APCs) promote glucose uptake by upregulating the membrane expression of the glucose transporter Glut1 through the mammalian target of rapamycin (mTOR) pathway [5,6,7]. These effects are counteracted by immune inhibitory molecules, including CTLA4 and PD-1, that brake immune responses to a halt by inhibiting glucose uptake and metabolism as well as by promoting endogenous fatty acid oxidation (FAO) [8].

FAO and mitochondria play important roles in both effector and memory T cells’ metabolism [9]. Mitochondria are essential for the development of memory responses since remodeling of these organelles is required to maintain adequate levels of FAO that allow memory T cell survival. Indeed, effector and memory T cells feature distinct mitochondrial patterns underlying the different engagement of FAO and OXPHOS in these activation states. The enhanced aerobic glycolysis rate upon naive T cells activation is accompanied in effector T cells by mitochondria fission that leads to cristae expansion and thus to a reduction in the electron transport chain (ETC) efficiency. On the contrary, memory T cells feature networks of fused mitochondria that tighten cristae and closely associate ETC complexes, ultimately fostering FAO and OXPHOS [10].

More recently, a role for cardiolipins, a class of phospholipids specifically associated with the internal mitochondrial membrane, has been uncovered in CD8 T cell activation and differentiation into memory cells. CD8 T cells deficient for the cardiolipin-synthesizing enzyme PTMPT1 (protein tyrosine phosphatase mitochondrial 1) show defective responses to antigens in vivo both at the induction and at the reactivation phase and fail to achieve complete pathogen clearance. Similarly, the genetic deficiency of Tafazzin, an enzyme involved in cardiolipin remodeling (as a protective mechanism from oxidative damage), also results in CD8 T cells with reduced proliferative ability and functionality and in a reduced number of central memory CD8 T cells in vivo [11].

## 3. A Closer Look at the Metabolism of Tregs

Regulatory T cells (Tregs) are now recognized as a specialized CD4 T cell subset essential for immune homeostasis, as well as for protection from autoimmunity and excessive inflammation. Although originally described as anergic cells [12], Tregs can be highly proliferative, especially in specific circumstances in vivo, such as the expansion occurring in neonatal life [13,14] or in tissue injury and cancer [15,16].

The metabolic dynamics in Tregs have not yet been completely clarified, most likely because of the intrinsic heterogeneity and plasticity that characterizes this immune subset. For instance, the differentiation of Tregs from conventional CD4 T cells (Tconvs) in the periphery seems to occur when the glycolysis–lipogenesis pathway is blocked. Accordingly, in vitro differentiation of Tregs from Tconvs can be induced by rapamycin, a well-known inhibitor of mTOR [17]. Nevertheless, additional studies on natural Tregs have demonstrated that the mTOR-driven glycolytic–lipogenic metabolism plays a pivotal role in Treg expansion and fitness. Indeed, mice carrying the Treg-intrinsic ablation of mTOR spontaneously develop a severe autoimmune and inflammatory systemic disease [18,19], secondary to a decreased glycolytic rate and thus reduced competitive fitness at the systemic level in Tregs [18].

An additional factor affecting Tregs metabolism is probably the activity at which they are analyzed. Indeed, many studies suggest that some “division of labor” exists between different metabolic pathways in Tregs, with glycolysis mainly involved in migration and OXPHOS in suppressive functions [20]. The engagement of pro-migratory stimuli, such as lymphocyte function-associated antigen 1 (LFA-1) or CD28, enhances the glucose uptake and glycolytic rate while promoting migration through the activation of mTORC2 and, ultimately, the hexokinase isoenzyme glucokinase (GCK) [21].

Accumulating data have linked mitochondrial activity, and thus OXPHOS, to Treg-suppressive functions. In a recent report, mice with Treg-restricted ablation of the mitochondrial complex III developed a severe systemic disease, characterized by apparently normal Treg numbers but defective Treg competitive fitness and Treg suppression at the systemic level [22]. Other genes related to mitochondria functions, including the complex I of electron transport, have been recognized as important elements for Treg suppressive function [23,24]. Besides fueling Treg functions, the proficient mitochondrial metabolism may control their suppressive functions through mechanisms not strictly related to OXPHOS. In a recent paper, Field et al. have proposed that mitochondria integrity represents a checkpoint for Treg suppressive activity. Specifically, the Treg-restricted deficiency of the fatty acid transporter FABP5 results in enhanced suppressive functions by promoting IL-10 release. The underlying mechanism involves the release of mtDNA in the cytoplasm that activates IFN-I-related genes through the cyclic AMP-GMP synthase-Stimulator of interferon genes (cGAS-STING) pathway. Interestingly, the phenotype of FABP5-deficient Tregs is recapitulated in the tumor microenvironment by the reduced lipid availability. An IFN-I-related signature is described in tumor-infiltrating Tregs, suggesting that mitochondrial damage may exacerbate Treg suppressive functions in this setting [25].

## 4. Analysis of Mitochondrial Metabolism in T Cells: Role in the Tumor Microenvironment

It is now well established that mitochondrial morphology and functions play crucial roles in a variety of immune functions and immune cell types. Along T cell differentiation, mitochondria undergo extensive remodeling that is tightly intermingled with functional reprogramming. Memory CD8 T cells have a higher spare respiratory capacity than effector and naive cells, and this is associated with higher mitochondrial mass and fatty acid oxidation [26]. Moreover, memory cells display mitochondria with fused morphology, and conversely mitochondrial fusion supports memory T cell development [10]. Costimulation through the CD28 during T cell priming is the key event that drives fatty acid oxidation, mitochondrial elongation and respiratory capacity in developing memory T cells [9].

Within the tumor microenvironment, T cells progressively lose their effector functions undergoing metabolic and epigenetic reprogramming, a process known as exhaustion [27]. Several studies demonstrate that mitochondrial alterations significantly contribute to tumor-associated T cell exhaustion. In mouse tumor models, tumor-infiltrating CD8 T cells display an accumulation of depolarized mitochondria that are not efficiently cleared because of defective mitophagy; these cells have functional impairment and epigenetic reprogramming that are compatible with exhaustion [28,29]. The balance between mitochondrial fusion and fission may dictate the fate of T cell activity: supporting this view, the manipulation of mitochondrial pro-fission (Drp1) or pro-fusion (Opa1) proteins can modulate T cell effector functions and affect their exhaustion in the tumor microenvironment [10,30]. Whether CD4 T cell activation and effector functions are similarly affected by mitochondrial dynamics is less clear. In a mouse model of experimental autoimmune encephalomyelitis, inhibiting mitochondrial fission suppressed T helper 1 and 17 polarization, thus alleviating disease severity [31]: these data support the idea that mitochondrial fission is required for CD4 T cell effector functions, similarly to CD8 T cells.

Several factors are recognized to drive mitochondrial dysfunctions in tumor-infiltrating lymphocytes (TILs). In human cancer, the tumor microenvironment accumulates long-chain fatty acids that are incorporated (and not metabolized) by CD8 T cells: this drives functional impairment and compromises mitochondrial integrity and polarization, probably due to the disruption of mitochondria-associated ER membrane that is induced by palmitate overload [32]. However, not all T cells are equal. Indeed, in another experimental setting, fatty acid uptake by Tregs through the transporter CD36 was rather required for their suppressive function in tumors, and also for proper mitochondrial biogenesis and oxidative phosphorylation [33]. Recent data have highlighted the role of systemic metabolism in shaping local immunometabolic responses in tumors: indeed, obesity leads to increased fatty acid uptake by tumor cells but not by T cells, promoting their functional exhaustion [34]. Hypoxia in the tumor microenvironment is another factor that may play a key role in T cell exhaustion: indeed, severe hypoxia drives mitochondrial dysfunctions and ROS formation, and this promotes the development of an exhausted phenotype [35].

The information about mitochondrial activities in T cell functions has been growing in the latest years. This advance would not be possible without the usage of staining techniques at the single cell level. Besides electron microscopy that allows a detailed analysis of mitochondria ultrastructure, several fluorescence-based staining techniques have been developed to study these organelles and assess their functional state. Maintenance of the proper potential across the inner and outer membranes allows mitochondria to fulfil their tasks [36]; therefore, membrane depolarization is a good indicator of mitochondrial dysfunction [37,38].

The available staining techniques identify cells with healthy, metabolically active mitochondria through fluorescent reagents that accumulate in active mitochondria with intact membrane potentials and are suitable both for imaging microscopy and flow cytometry. These fluorescent probes can be classified into two main categories according to their ability to detect the mitochondrial membrane potential at a specific time point or its dynamic changes over time.

The first category includes the MitoTracker probes, small (<1 kDa) cell-permeant mitochondrion-selective dyes containing a thiol-reactive chloromethyl moiety that forms covalent bonds with the mitochondrial thiols. As a consequence of these irreversible reactions, MitoTracker probes can only be used to detect mitochondrial membrane potential at the time of loading in live cells, with active mitochondria exhibiting brighter fluorescence compared to apoptotic mitochondria. The second category includes fluorescent dyes that show potential-dependent accumulation in mitochondria and change their emission properties upon depolarization, thus allowing for the monitoring of the potential. Some assays are based on dual-emission dyes for a ratiometric assessment of the membrane depolarization. For instance, when the JC-1 (tetraethyl benzimidazolyl carbocyanine iodide) dye accumulation within the mitochondria is high enough in concentration J-aggregates form, and there is an emission shift from the monomer (green, ~529 nm) that is prominent at lower dye concentrations to J- aggregates (red, ~590 nm) which are formed as the dye concentration increases. Consequently, mitochondrial depolarization is indicated by a decrease in the red/green fluorescence intensity ratio. With a similar mechanism, the DiOC2(3) (3,3′-Diethyloxacarbocyanine iodide) shifts from the green to the far-red emission (above 650 nm). Dynamic changes of mitochondrial membrane potential can be monitored by single emission molecules, such as (tetramethylrhodamine methyl ester) (TMRM) and DiIC1(5). TMRM (tetramethylrhodamine methyl ester) or the related TMRE (tetramethylrhodamine ethyl ester) are small, cell-permeant dyes that accumulate in active mitochondria. If the cells are healthy and have functioning mitochondria, the signal is bright. Upon the loss of the mitochondrial membrane potential, TMRM and TMRE accumulation ceases and the signal dims or is lost from the mitochondria.

## 5. Control of T Cell Metabolism at the Transcriptional Level

T cell activation through the TCR and costimulatory and cytokine receptors initiates a cascade of phosphorylation and calcium flux events that finally end in the activation of transcription factors and thus in gene expression. This signaling pathway concomitantly induces the activation of mTOR: this factor translates the immune stimulation into a metabolic switch through a top–down pathway. Conversely, several metabolites have been shown to affect immune signals, and this can be seen as a bottom–up pathway [2]. In both cases, the modulation of immune and metabolic T cell functions may imply de novo gene expression: indeed, most of these metabolic sensors act through a combination of post-translational, post-transcriptional and transcriptional events. Therefore, studying gene expression can provide significant insights into the metabolic status in which T cells enter under specific conditions.

As the major signal in the top–down pathway, mTOR drives a phosphorylation cascade that ends in the activation of key transcription factors, such as c-Myc, sterol regulatory element binding proteins (SREPBs) and hypoxia inducible factor 1 alpha (HIF1), through which mTOR prompts lipid biosynthesis and glucose metabolism [39]. Besides these metabolic players, mTOR directly and indirectly controls a wide array of transcription factors that are involved in T cell development, activation, polarization or differentiation, such as Nuclear Factor-κB (NF-κB,) Forkhead box O (FOXO), and Signal Transducer and Activator of Transcription (STAT) families [40]. This implies that the metabolic activation of T cells, driven by mTOR, translates in a complex transcriptional reprogramming.

Following TCR and CD28 engagement, c-Myc is induced and maintained through both transcriptional and post-transcriptional mechanisms, also supported by mTOR [39]. C-Myc is strictly required for cell growth and division: indeed, it induces a metabolic program based on the transcription of genes related to glucose catabolism, glutaminolysis and polyamine biosynthesis [41]. However, further layers of c-Myc signaling became apparent when the proteome and the phosphoproteome of activated T cells were analyzed: this study revealed the induction of certain c-Myc targets that were involved in ribosome biogenesis and translation initiation, thus pointing to post-transcriptional regulation by c-Myc of T cell activation [42].

Sterol regulatory element binding proteins (SREPBs) are transcription factors that promote the expression of genes devoted to fatty acid and cholesterol biosynthesis. They are activated, in an mTOR-dependent fashion, in T cells that receive a mitogenic stimulus, and are responsible for the lipid anabolism required for T cell division [43]. Interestingly, SREPBs are sensitive to the intracellular sterol content and initiate their transcriptional program under conditions of sterol deprivation [44], thus they may integrate mitogenic signals with the context of nutrient availability along T cell activation.

Hypoxia inducible factor 1 alpha (HIF1α) is a transcription factor that senses low oxygen tension and translates this environmental message into the switch from oxidative to glycolytic metabolism [45]. Several key roles have been identified for HIF1α in T cell functions: as an example, HIF1α drives T helper 17 cell polarization at the expense of Treg conversion, and this event is mediated by its direct transcriptional induction of the IL-17 gene [46]. Notably, HIF1α-deficient T cells showed the marked repression of a series of metabolism-related genes, and especially those involved in glycolysis [47].

Several metabolites can affect T cell functions in a bottom–up direction: for instance, the AMP/ATP ratio regulates Adenosine Monophosphate-activated Protein Kinase (AMPK) activity, some amino acids work as mTOR activators through several processes, and membrane phospholipids and cholesterol affect TCR signaling [2]. Some of these bottom–up events can occur through transcriptional or post-transcriptional regulation. Indeed, there are several examples of metabolic enzymes that display “moonlighting” functions, in that they are employed in gene regulatory networks when not engaged in their metabolic cascade. Glyceraldehyde 3-phosphate dehydrogenase (GAPDH) catalyzes the sixth step of glycolysis; interestingly, it is one the most widely used among housekeeping genes or proteins in expression experiments. In several cell types, GAPDH has been shown to bind certain mRNAs and to repress translation; in T cells, this activity is exerted on the 3′ untranslated region (UTR) of *Ifng* mRNA, implying that when glycolysis is sustained, the IFNγ protein can be translated, and that glycolysis and effector function are mechanistically linked through GAPDH [48]. Similarly, enolase 1, another enzyme of the glycolytic cascade, controls the transcription of certain splicing variants of the human FOXP3 gene, when it is not engaged in glycolysis [49].

Another level of the bottom–up regulation exerted by metabolic signals is the epigenetic remodeling. Indeed, the extent of histone and DNA acetylation and methylation can be profoundly affected by the availability of their substrates, namely acetyl and methyl groups. The citrate produced during the TCA cycle is exported through specific carriers from mitochondria to cytosol, where the enzyme ATP citrate lyase regenerates Acetyl-CoA, a major donor of acetyl groups for histone acetylation. In T helper 1 cells, this metabolic pathway is crucially required for histone acetylation and the transcription of genes related to effector function such as IFNγ [50]. Other intermediates of the TCA cycle can affect methylation, acting as cofactors or inhibitors of specific demethylases. In Tregs, the genetic deficiency of mitochondrial complex III leads to a complete loss of Treg functions, in association with increased DNA methylation and with higher levels of 2-hydroxyglutarate and succinate, two metabolites with the known ability to suppress DNA demethylases [22]. Metabolic and epigenetic remodeling are tightly linked in the establishment of T cell exhaustion [27]; as an example, glutamine blockade rescues anti-tumor T cell effector functions while promoting the replenishment of the TCA cycle [51], which provides intermediates with the ability to modulate demethylase activity.

## 6. RNA Flow Cytometry: Pros and Cons Compared to Other Techniques

By virtue of the tight crosstalk between metabolism and transcriptional/post-transcriptional regulation, a quantitative analysis of gene and/or protein expression may provide relevant information on the activation of specific metabolic pathways. In line with this idea, Ahl et al. have recently developed a flow cytometry-based metabolic profiling of human leukocytes, named Met-Flow, which relies on the intracellular staining and multiparameter analysis of several key and rate-limiting metabolic proteins [52]. This approach allowed one to estimate which metabolic pathways were dominant among different leukocyte subtypes, or after in vitro activation, or following selective pharmacological inhibition. For the first time, flow cytometry allowed one to dissect major metabolic pathways in a heterogeneous population at the single-cell level and at the protein level. However, the lack of reliable antibodies covering the majority of intracellular enzymes, and the combination with intranuclear markers that may be required to identify certain cell types, may hinder the expansion of this method.

As compared to proteins, nucleic acids show a simpler molecular structure that renders the design of detection probes very straightforward. Indeed, at least theoretically, a specific gene locus or RNA molecules of a known sequence can be detected by one or more complementary probes. The gene expression dynamics underlying T cell metabolism render RNA analysis an essential tool for immunological studies in the field. Standard methods for RNA analysis currently include quantitative and real-time PCR, expression microarrays and RNA sequencing for the transcriptome analysis [53]. These techniques show high sensitivity, are suitable for high-throughput experiments and allow population- or single cell-restricted analysis upon sample sorting and represent a gold standard for accurate RNA quantification. Immunological studies often involve experimental conditions with rare cell populations or limited sample amounts that render cell sorting impractical and sometimes not feasible. In the last decade, flow cytometry-based approaches have been developed that allow single-cell RNA detection from heterogeneous samples, that include the SmartFlare ^TM^ (EMD Millipore, Burlington, MA, USA) system and the Prime Flow ^TM^ (Thermo Fisher Scientific, Waltham, MA, USA). As compared to earlier detection attempts, such as the fluorescent in situ hybridization (FISH), these methods preserve better cells and fluorochromes properties, resulting in better sample quality. Moreover, unlike FISH and other microscopy based-methods, they are also feasible for non-adherent cells staining and allow high-throughput multiparametric screening because RNA detection can be carried out in association with phenotypical analysis [54,55].

The SmartFlare technology takes advantage of a three-component system, where oligonucleotides complementary to a specific RNA are conjugated to gold nanoparticles and bound to a fluorescent-tagged reporter (Figure 1). The conjugate is internalized into the cytoplasm and the RNA of interest, if present inside the cell, will displace the lower affinity fluorescently tagged reporter from the complementary oligonucleotide. Consequently, the reporter oligonucleotide is no longer quenched, and the emitted fluorescence can be detected by a flow cytometer. Although the SmartFlare™ does not affect cell viability and is thus feasible for downstream experiments, it has not met a broad application for multiparametric immunological studies and the limited availability of fluorophores (only Cy3 and Cy5) is probably among the reasons. Nevertheless, the technology has proven useful for the characterization of tumor cell lines, screening the efficacy of new drugs [56] and in the identification of pluripotent stem cells in different mammalian tissues [57].

The multiparametric setting of immunological studies, especially those focused on metabolism, have taken much advantage of RNA analysis based on the branched DNA technology, namely Prime Flow^TM^ and the strictly related FISH-flow (Table 1). Prime Flow^TM^ allows for the simultaneous detection of up to four target RNA of different types (mRNA, microRNA, viral and long non-coding RNA) in combination with surface and intracellular/intranuclear staining. In the assay workflow, cells that have been previously stained with antibodies specific to surface molecules are fixed and permeabilized using a proprietary buffer that preserves RNA transcripts. Subsequently, a hybridization step is performed with oligonucleotide pairs, or target probes (TBs), that hybridize on complementary sequences on the target RNA (Figure 1). Each batch contains 20–40 TB pairs (or a single pair in the case of miRNA probe sets) to allow both for specificity and for signal amplification. Indeed, only if all the TB pairs hybridize adjacent to each other the subsequent recognition by pre-amplifier and amplifier oligonucleotides can occur. After the amplification steps, oligonucleotides conjugated to fluorochromes (label probes) are added to hybridize with multiple sites on the amplifiers. An optimally assembled branched DNA complex can provide a theoretical 8000- to 16,000-fold signal amplification of the targeted RNA [54].

The different types of RNA detectable by Prime Flow result in a broad range of applications that are summarized in Table 1. The simultaneous detection of RNAs and their cognate proteins allow one to study their dynamics in specific cell populations. For instance, the combination of Tumor Necrosis Factor (TNF) α mRNA and cognate protein detection in lipopolysaccharide (LPS)-stimulated macrophages at different time points has been used to identify the length of incubation needed to initiate protein translation [103]. Post-transcriptional regulation can be more finely studied in specific cell types by combining protein expression and/or mRNA with the detection of microRNA. Lai and coworkers unveiled unique microRNA profiles in the four peritoneal cell subsets of murine macrophages by combining microRNA, mRNA and protein detection [118].

RNA flow cytometry also offers an alternative detection tool when the measurement of a protein is technically infeasible either because no quality antibody exists, or the epitope is not accessible [54]. We have used Prime Flow ^TM^ to study genes involved in fatty acid metabolism (*Pparg*, *Acacb*, and *Cpt1a*) in intratumoral conventional CD4 T cells and Tregs [83]. Besides overcoming the antibody availability, we were able to perform the analysis on single tumors, thus taking into account sample heterogeneity. Indeed, the isolation of tumor-infiltrating lymphocytes by single cell sorting in mice is often performed on pools of several samples to achieve cell numbers adequate for gene or protein expression analysis, but to the disadvantage of accuracy. Another noticeable application of FISH-flow has been the identification of latent HIV infection in peripheral blood cells. The combined staining with *Gagpol* mRNA and the Gag protein allowed the identification of virus infected CD4+ T cells in patients undergoing antiretroviral therapy with a detection limit of *Gagpol* mRNA+/Gag protein+ cells per million CD4+ T cells [110].

## 7. An Example of Combined Detection of RNA, Nuclear Factors and Mitochondria by Flow Cytometry

In the latest years, the interest in defining a complete profile of tumor-infiltrating immune cells by a phenotypic, transcriptional and metabolic point of view has become noticeable, especially considering the impact of “immune contexture” in modulating cancer progression and in determining patient prognosis [121]. A substantial field of study focused on the characterization of Tregs in the cancer microenvironment, since these cells are thought to be a major barrier to therapeutic efforts to mobilize the immune system against the tumor [122]. The possibility of identifying unique characteristics that could distinguish tumor-infiltrating Tregs from other Tregs in the body, as well as from the beneficial anti-tumor effector T cells within tumors, could be exploited to target or to reprogram tumor Tregs for cancer treatment [123]. A good knowledge about Tregs has been achieved regarding specific aspects: (i) the existence of a transcriptional signature has been highlighted, specifically associated with Tregs in thousands of tumors from different stages and locations and conserved across species [124]; (ii) the expression of a miRNA in a subset of tumor Tregs has been reported to regulate their proliferation and immunosuppressive capacity [125]; (iii) many metabolic peculiarities have been associated to specific biological aspects of Tregs in tumor microenvironment [20], and in this context, a key role for Treg suppressive function has been attributed to molecules that regulate mitochondrial integrity and function [25].

The technical ability to recognize and intersect as many features as possible, and to simultaneously compare different kinds of data (including RNA expression and metabolic traits) from several cell populations interacting in the same heterogeneous tissue, such as a tumor, could be crucial to highlight differences potentially useful for focused therapeutic intervention. Several examples can be found in the literature of the usage of the Prime Flow technique to analyze T cells in complex samples. Thanks to this approach, we had the possibility to associate the phenotypic profile of tumor-infiltrating lymphocytes (TILs) to the expression of key genes of specific metabolic pathways, and these data were confirmed by a gene expression analysis [83]. Others could unveil the critical role of a stress-responsive transcription factor in the maintenance of TIL mitochondrial fitness and function by using the respective RNA-labelled probe and could discriminate a differential expression of this gene in phenotypically, functionally and metabolically different TIL subsets [126]. Another group identified the high expression of the adenosine receptor A2AR in central memory CD8 T cell subset through its mRNA probe detection: this receptor dictated the TIL susceptibility to adenosine-mediated suppression in the tumor microenvironment that impairs CD8 T cell metabolic fitness and cytokine production [69]. However, especially in the latest two works, the overall significance of the data resulted from the contribution of different analysis techniques and not only Prime Flow analysis.

Here, we describe a protocol that combines the identification of TIL phenotypical characteristics with RNA detection by Prime Flow assay and with the measurement of mitochondrial function by using a specific fixable dye, belonging to the MitoTracker family dyes, named MitoTracker Deep Red (MDR). This protocol allows one to evaluate all these aspects in Tregs and CD4+ conventional T cells (Tconvs) in the same sample by flow cytometry. The subcutaneous-implanted colon carcinoma cell line MCA38 was used as a tumor model. TILs were enriched through a 40/80 Percoll density gradient from tumor samples, while splenocytes were used as the control. The staining protocol is outlined in the workflow in Figure 2.

In more detail, at first cells were incubated with the fixable viability dye (APC-eFluor780) to allow for the detection and exclusion of dead cells, and then the staining for surface antigens was performed by using the following antibodies: CD4 (Brilliant Violet 785) for cell lineage identification, CD44 (Brilliant Violet 510) as activation marker, OX40 (Brilliant Violet 605), a member of TNF-receptor superfamily highly expressed in tumor Tregs and CD71 (Brilliant Violet 421), the transferrin receptor. The latter two markers were analyzed based on data in the literature and data from our laboratory: genes encoding for OX40 and CD71 belong to a conserved tumor-specific Treg signature that has been identified in a meta-analysis by Zheng et al. [127]; moreover, we demonstrated that OX40 expression is distinctive of a proliferating, stable and highly suppressive activated Treg subset in tumor tissue [128,129].

After surface staining, cells were incubated with MDR, a carbocyanine-based and far red-fluorescent dye (absorption/emission ~644/665 nm), that passively diffuses across the plasma membrane and stains mitochondria in live cells, allowing for the quantification of the active mitochondrial mass [130]. Different cell-permeant MitoTracker probes for mitochondria labelling exist which differ in spectral characteristics, fixability and fluorescence mechanism. In this experimental setting, MDR was chosen because it is retained during cell permeabilization and fixation (required for intracellular staining and Prime Flow assay) and it is suited for multicolor labelling experiments.

Then, we started the Prime Flow protocol: cells underwent the first of two fixation steps, in order to warrant the immobilization and stability of the surface and intracellular antigens, as well as the mitochondrial labelling and target RNA. After this fixation step, cells were permeabilized by two washings with a permeabilization buffer containing RNase inhibitors, in order to preserve the integrity of mRNA transcripts, and then intracellular staining for Foxp3 (PE-eFluor610), the master regulator gene of Tregs, and Ki67 (PE-Cy7), a nuclear marker of cell proliferation, was performed.

At this point, cells underwent the second fixation step, followed by the first hybridization step, which allows for the annealing of mRNA-specific oligonucleotide “target probe” to the RNA sequence of interest, β-Actin in our case. Beta-actin is generally used as a housekeeping gene in several techniques. After the specific mRNA targeting, sequential hybridization steps were performed, based on branched DNA chemistry, in order to increase the specificity of the detection system and to greatly amplify the signal of annealed target probes. At the end of the procedure, labelled samples were acquired on a cell analyzer and were analyzed with appropriate software.

As a result of this optimized protocol, we had the opportunity to measure the expression of transcripts (β-Actin) together with surface and nuclear antigens at the single cell level, in combination with a dye that gave us information about the cell metabolic status. Interestingly, we could compare all these aspects, simultaneously, in distinct cell populations that inhabit the same heterogeneous tissue. This analysis allowed one to unveil that, compared to Tconvs, and to Tregs from spleen, tumor-infiltrating Tregs expressed a higher level of activation markers (CD44, CD71 and OX40), proliferated at a higher extent (as indicated by the higher Ki67 expression) and showed an increased mitochondrial mass (as revealed by MDR staining). As expected, all the cells expressed high levels of β-Actin mRNA (Figure 3).

These results confirm previous data [83] indicating that tumor-infiltrating Tregs have a profile that is compatible with functional and metabolic activation. Indeed, compared to spleen-Tregs and to Tconvs, they express very high levels of OX40 and also contain a relevant proportion of proliferating cells. This is in line with previous studies showing a proliferative role of OX40 on Tregs [131]. This phenotype was not associated with a significant increase in mitochondrial mass, as revealed by MDR staining. Others have shown that, in a different tumor model, tumor-infiltrating Tregs contain a lower mitochondrial mass compared to spleen Tregs [25]: it is thus possible that mitochondrial biogenesis is not required for the activation and functions of Tregs at the tumor site.

The multiparameter flow cytometry data obtained from this type of experiment were suitable for multidimensionality reduction and clustering analysis using bioinformatic tools such as t-distributed Stochastic Neighbor Embedding (tSNE) and/or Uniform Manifold Approximation and Projection (UMAP) (Figure 4). To perform this analysis, CD4 T cells from spleen and tumors were gated, downsampled and concatenated together, then, Tregs and Tconvs were manually identified based on Foxp3 expression (Figure 4A). Both UMAP and tSNE analyses displayed that Tregs and Tconvs from tumors and spleens populated distinct areas in the bidimensional plots (Figure 4B). Interestingly, the visualization of marker coexpression highlighted that proliferating (Ki67+) cells tended to display distinctive features (Figure 4C). Interestingly, proliferating cells were characterized by higher levels of β-Actin mRNA expression: this observation suggests that β-Actin analysis may not only be useful as a positive control for the RNA flow assay but also give potential information on T cell activation. Indeed, TCR-dependent reorganization of the actin cytoskeleton is a key event in the formation of the immunological synapse and in cell motility [132], and our data suggest that the control of β-Actin activity may also be exerted at the transcriptional level in proliferating cells.

## 8. Conclusions

Metabolic switches are key events and dictate the outcome of T cell activation, thus their characterization is of utmost importance in understanding the T cell response to microenvironmental signals and nutrients. Studying gene expression can provide significant information on the main metabolic pathways that are active in specific cell types and states. Recent advances in flow cytometry allow for the simultaneous analysis of specific mRNAs and metabolic, phenotypic and functional markers in complex populations, with unprecedented resolution. This approach may be instrumental in studying how T cells adapt their metabolism and thus their functions to cope with metabolic restriction in certain tissue microenvironments, such as the tumor site.

Besides contributing to produce high quality data for basic research, this approach also has potential clinical relevance. For instance, the reactivation of effector T cells in cancer patients undergoing immunotherapy with immune checkpoint inhibitors relies on the modulation of metabolic pathways in exhausted cells [27]. In this view, monitoring the metabolic status of effector lymphocytes in these patients might be useful to assess their response to therapy in real time. Indeed, RNA flow cytometry can be performed on samples of limited size (e.g., peripheral blood leucocytes from routine blood tests) as compared to classical metabolic assays ex vivo and in a relatively short time. Moreover, as we discussed in the text, the study of metabolism can be performed in parallel with phenotypical and miRNA analysis that allows for correlations with activation, proliferative status of cells and possibly post-transcriptional regulation. Another potential clinical application we envisage for RNA flow includes the identification of rare viral reservoirs in patients chronically infected with HIV as well as other potentially harmful pathogens.

## Figures and Tables

**Figure 1 ijms-22-03906-f001:**
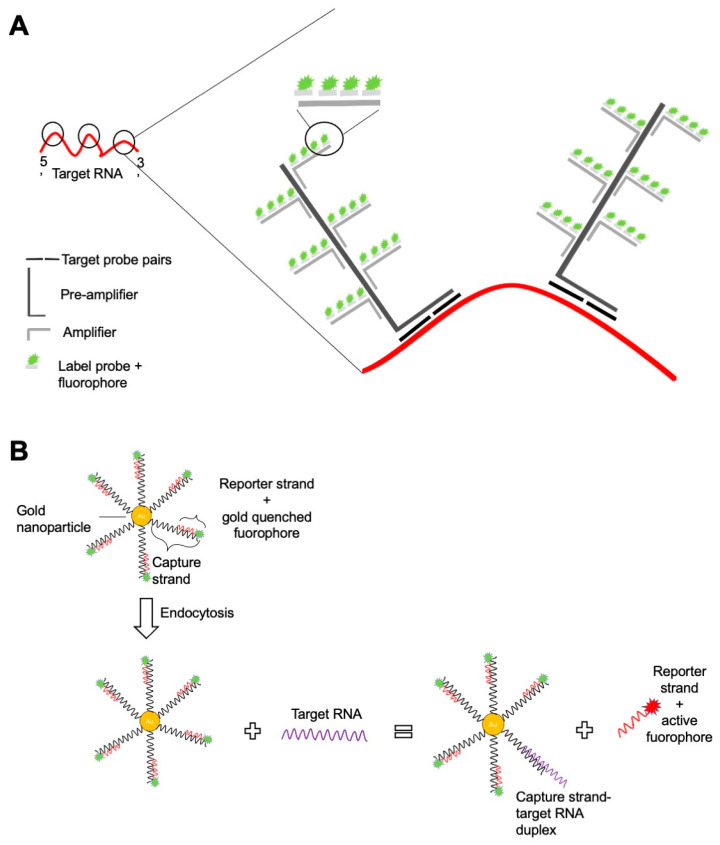
Scheme of the branched-DNA chemistry applied in the Prime Flow ^TM^ protocol (**A**) and of the SmartFlare chemistry (**B**).

**Figure 2 ijms-22-03906-f002:**
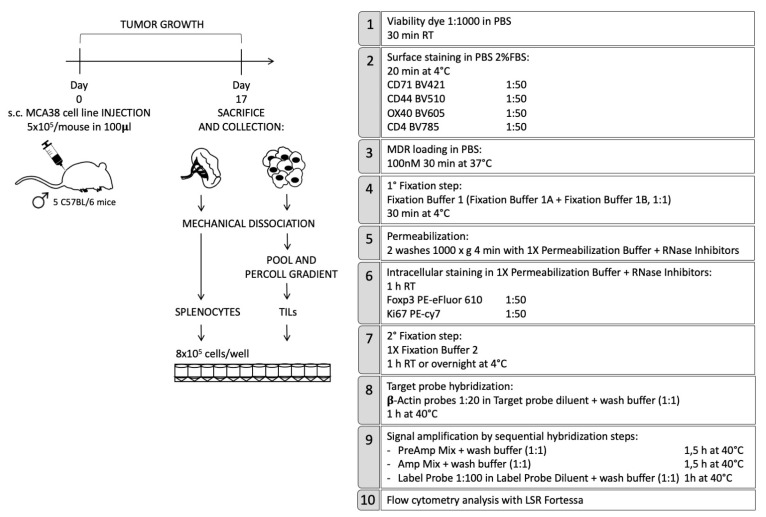
Workflow of the combined staining of surface and nuclear markers, mitochondria and specific mRNAs. PBS, phosphate buffered saline.

**Figure 3 ijms-22-03906-f003:**
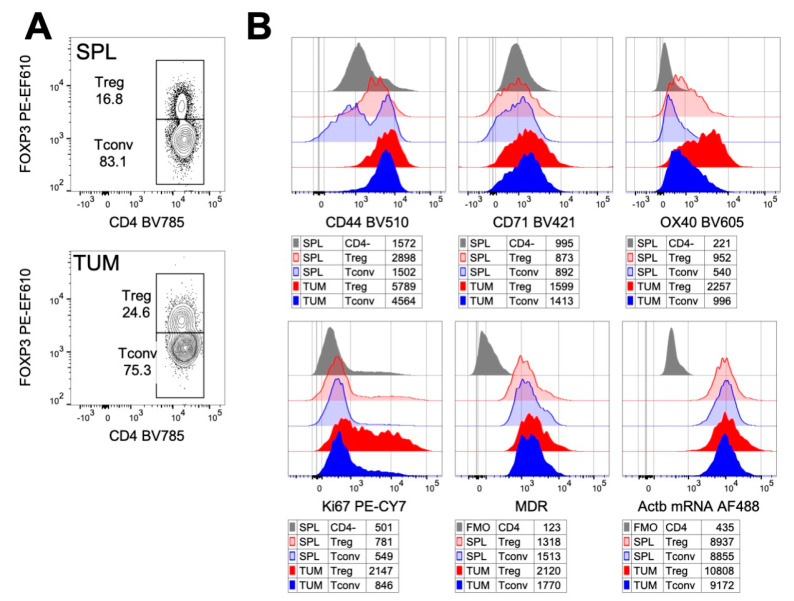
Detection of β-Actin mRNA in combination with surface and nuclear proteins and with mitochondrial mass in Tregs and Tconvs from splenocytes (SPL) and tumor tissue (TUM) by flow cytometry. MC38 cells (5 × 10^5^) were subcutaneously injected into C57BL/6 mice and Prime Flow assay and flow cytometry analysis were performed on lymphocytes extracted from splenocytes and tumor tissue at 17 days post-tumor transplantation. Staining was performed according to the workflow in Figure 2. (**A**) Contour plots showing the strategy for the identification of Tregs and Tconvs in SPL and TUM, according to Foxp3 intranuclear expression. (**B**) Overlay of the histograms showing the expression of each marker in the indicated cell subset and tissue. Numbers in the legends indicate the geometric mean fluorescence intensity (gMFI). Grey histograms represent the fluorescence-minus-one (FMO) negative controls or CD4-negative cells, as control.

**Figure 4 ijms-22-03906-f004:**
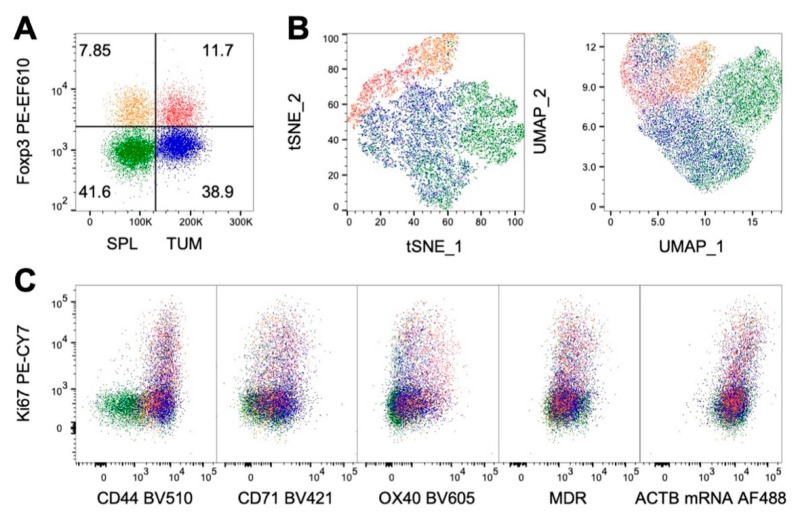
Unsupervised clustering analysis of flow cytometry data obtained from Tregs and Tconvs from spleen (SPL) or tumors (TUM). Gated CD4 T cells were downsampled and concatenated together and Tregs and Tconvs in both samples were manually identified according to Foxp3 expression. (**A**) Plot showing the overlay of gated Tregs and Tconvs in concatenated CD4 T cells from SPL and TUM. (**B**) tSNE (left) and UMAP (right) visualization of the distribution of Treg and Tconv populations from SPL and TUM in the concatenated samples according to the expression of markers indicated in Figure 3. Both multidimensionality reduction analyses were run with default settings in Flowjo 10.5.3. (**C**) Representation of the co-expression of Ki67 with the other markers in overlayed Treg and Tconv subsets in concatenated CD4 T cells from SPL e TUM samples.

**Table 1 ijms-22-03906-t001:** RNA and protein analysis by flow cytometry.

Technique	Molecular Target	Platform	Application	Refs
Met-Flow	Proteins	Flow cytometry: BD X-30 FACSymphony (27 colors)	Characterization metabolic pathways in immune cells	[52]
SmartFlare	RNA	Flow cytometry Fluorescence plate reader	Studying gene expression in live cells for downstream application	[56,57,58,59,60,61]
PrimeFlow FISH-Flow	RNA	Flow cytometry: BD LSR Fortessa (13 colors), BD LSR II (2 to 11 colors), Beckman Coulter Gallios (6 to 10 colors)	Analyze mRNA expression at the single-cell level	[62,63,64,65,66,67,68,69,70,71,72,73,74,75,76,77,78,79,80,81,82,83]
			Analyze RNA and protein kinetics in the same cell	[84,85,86,87,88,89,90,91,92,93,94,95,96,97,98,99,100,101,102,103,104]
			Detection of viral RNA in infected cells	[105,106,107,108,109,110,111,112,113,114]
			Detect microRNA (miRNA)	[90,115,116,117,118,119,120]

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
