# Peer review of "RNA Flow Cytometry for the Study of T Cell Metabolism"

_ijms, 2021, doi:10.3390/ijms22083906_

Round 1

Reviewer 1 Report

Rossi et al describe the importance of using RNA flow cytometry for the study of T cell metabolism. Their review provides some important perspective of an emerging technology that can provide important multiparameter data in heterologous environments. While the review is well written overall, there are a couple important points to address.

  1. The manuscript title and abstract talk about RNA flow, but the review does not mention flow cytometry until the 5th of 7 sections. It would be good to start out the review by explaining why RNA flow is useful for measuring metabolic function before jumping into 4 sections of T cell metabolism. This could be done with a couple of sentences similar to those found in the conclusion section.
  2. Section 3 (lines 117-127) discusses in multiple situations how CD8 T cell mitochondria fusion and fission are key parameters. It would be good to include if this is also the case for CD4 T cells.
  3. Section 3 (lines 128-131) states that long chain fatty acids in the tumor environment impairs mitochondrial integrity and polarization. It would be good to include how this occurs.
  4. Section 3 (lines 177-179) contains a duplicate sentence that is also found at the beginning of section 2. It should be deleted in section 3.
  5. Section 5 jumps into RNA flow and mentions Met-Flow, SmartFlare TM, and Prime Flow TM. It would be helpful to explain the similarities and differences of these products. Why would one use one or the other? It would be ideal to include a table that summarizes the way each of these technologies are similar and different in function and applications.
  6. Most of the time was focused on Prime Flow, is this product able to do things better than SmartFlare or Met-Flow, or just a similar product that was less expensive?
  7. Please define exactly what is the TB pair is references on line 289.
  8. Section 5 ends with a statement that RNA flow is not as good for quantitative analysis of transcripts as qPCR is. It would be good to add a sentence describing exactly which applications RNA flow is ideally suited for.
  9. There is no discussion of what type of a flow cytometer is needed to perform RNA flow. It would be helpful to include the minimum channels or types of machines that are needed to do this type of an experiment. Certainly this is dependent on the number of markers that you want to look at and more channels is nicer, but can a 4 color or 6 color machine work fine for this type of work.
  10. Figure 1 would be improved by including how SmartFlare and Met-Flow work also work.

Author Response

Rossi et al describe the importance of using RNA flow cytometry for the study of T cell metabolism. Their review provides some important perspective of an emerging technology that can provide important multiparameter data in heterologous environments. While the review is well written overall, there are a couple important points to address.

We thank the reviewer for his/her comments.

1. The manuscript title and abstract talk about RNA flow, but the review does not mention flow cytometry until the 5th of 7 sections. It would be good to start out the review by explaining why RNA flow is useful for measuring metabolic function before jumping into 4 sections of T cell metabolism. This could be done with a couple of sentences similar to those found in the conclusion section.

We have added a new paragraph as Introduction, to better explain the rationale and the structure of the review.

2. Section 3 (lines 117-127) discusses in multiple situations how CD8 T cell mitochondria fusion and fission are key parameters. It would be good to include if this is also the case for CD4 T cells.

A sentence regarding CD4 T cells has been added where indicated (now section 4, lines 151-155).

3. Section 3 (lines 128-131) states that long chain fatty acids in the tumor environment impairs mitochondrial integrity and polarization. It would be good to include how this occurs.

An explanation of the underlying mechanisms has been added (lines 159-160).

4. Section 3 (lines 177-179) contains a duplicate sentence that is also found at the beginning of section 2. It should be deleted in section 3.

The duplicated sentence (now at lines 206-208) has been eliminated.

5. Section 5 jumps into RNA flow and mentions Met-Flow, SmartFlare TM, and Prime Flow TM. It would be helpful to explain the similarities and differences of these products. Why would one use one or the other? It would be ideal to include a table that summarizes the way each of these technologies are similar and different in function and applications.

A new Table 1 has been added, which summarises the main features of these three techniques and includes a list of relevant references.

6. Most of the time was focused on Prime Flow, is this product able to do things better than SmartFlare or Met-Flow, or just a similar product that was less expensive?

The differences between the three methods have been further clarified in section 6 (lines 297-300 and 317-334).

7. Please define exactly what is the TB pair is references on line 289.

A definition for TB has been added at line 338.

8. Section 5 ends with a statement that RNA flow is not as good for quantitative analysis of transcripts as qPCR is. It would be good to add a sentence describing exactly which applications RNA flow is ideally suited for.

The sentence at the end of section 5 (now 6) has been clarified. 

9. There is no discussion of what type of a flow cytometer is needed to perform RNA flow. It would be helpful to include the minimum channels or types of machines that are needed to do this type of an experiment. Certainly this is dependent on the number of markers that you want to look at and more channels is nicer, but can a 4 color or 6 color machine work fine for this type of work.

Some details about the type of suitable flow cytometers are now mentioned in the new Table 1.

10. Figure 1 would be improved by including how SmartFlare and Met-Flow work also work.

Figure 1 has been modified by adding a schematic representation of the SmartFlare system. We haven’t added a scheme of Met-flow since it is a classical intracellular staining.

Reviewer 2 Report

The authors provide a review of selective topics in T-cell and T-reg metabolism related to tumor-infiltrating T-cells and control mechanisms.  They also provide information about technical approaches for the detection of mitochondria and RNA transcripts by flow cytometry, in combination with surface markers.

Major critique

Unfortunately, it is not until section 6 (page 7) where the authors describe a flow cytometry protocol utilized in their research lab combining CD4, T-reg markers, a mitochrondrial marker and Ki-67, on a mouse model of tumor infiltrating T-cells. After a long build up to a potentially exciting flow cytometry tool, the data provided results in an anticlimactic and abruptly concluded manuscript, with no value shown for the assessment of mitochondrial mass (all T-cells look the same), no data on mRNA assessment (only beta-actin control shown), undecipherable conclusions regarding Ki-67, and t-SNE maps placed more for the show as they do not provide more information than the histograms. Despite these limitations, the manuscript appears very well written. If there is really a demonstrated potential for RNA flow cytometry for the study of T cell metabolism as the title claims (besides the invariable beta-actin control shown), the authors should focus the manuscript on describing data and conclusions by other groups using this specific technique.

Minor critiques

Line 177: Sentence fragment, please correct.

Line 417: Meaning of Tconvs should be mentioned, as T conventional(?)

Author Response

The authors provide a review of selective topics in T-cell and T-reg metabolism related to tumor-infiltrating T-cells and control mechanisms.  They also provide information about technical approaches for the detection of mitochondria and RNA transcripts by flow cytometry, in combination with surface markers.

Major critique

Unfortunately, it is not until section 6 (page 7) where the authors describe a flow cytometry protocol utilized in their research lab combining CD4, T-reg markers, a mitochrondrial marker and Ki-67, on a mouse model of tumor infiltrating T-cells. After a long build up to a potentially exciting flow cytometry tool, the data provided results in an anticlimactic and abruptly concluded manuscript, with no value shown for the assessment of mitochondrial mass (all T-cells look the same), no data on mRNA assessment (only beta-actin control shown), undecipherable conclusions regarding Ki-67, and t-SNE maps placed more for the show as they do not provide more information than the histograms.

In response to similar comments by other reviewers, we have added a novel Introduction paragraph that illustrates the general structure of our review. We would like to point out that we showed our flow cytometry data not because we aimed to conclude anything about Treg biology, but only with the purpose of displaying the feasibility of the procedure and the potential applications of this technique. For this reason, we decided to share data about the housekeeping gene and not another metabolic gene. Our goal here was not to draw conclusions on tumor-Treg phenotype, proliferation, mitochondrial mass, or other features. However, we have added some comments to the data in paragraph 7 (lines 471-479 and 499-505).

Despite these limitations, the manuscript appears very well written. If there is really a demonstrated potential for RNA flow cytometry for the study of T cell metabolism as the title claims (besides the invariable beta-actin control shown), the authors should focus the manuscript on describing data and conclusions by other groups using this specific technique.

We thank you the reviewer for his/her positive comments. We have added a new Table 1 that includes relevant studies using RNA Flow for different purposes. Specific examples are mentioned in the text (lines 360-362 and 374-378).

Minor critiques

Line 177: Sentence fragment, please correct.

The sentence fragment has been eliminated.

Line 417: Meaning of Tconvs should be mentioned, as T conventional(?)

The word “Tconvs” has been spelled out at first appearance (lines 96).

Reviewer 3 Report

The review titled “RNA flow cytometry for the study of T cell metabolism” by Silvia Piconese et al described the important role of T cell metabolism in T cell activation and differentiation, and can be well studied by RNA flow cytometry. The authors also established a protocol to detect surface and nuclear markers, mitochondria, and specific mRNAs. The review was well written, but I have one concern.
The authors used more than half of the length on T cell metabolism, but readers will expect more content on the "RNA flow cytometry". There are only three references[52-54] associated with RNA flow cytometry. More should be presented in section 5 or a separate section on the current state of RNA flow cytometry.

Author Response

The review titled “RNA flow cytometry for the study of T cell metabolism” by Silvia Piconese et al described the important role of T cell metabolism in T cell activation and differentiation, and can be well studied by RNA flow cytometry. The authors also established a protocol to detect surface and nuclear markers, mitochondria, and specific mRNAs. The review was well written, but I have one concern.
The authors used more than half of the length on T cell metabolism, but readers will expect more content on the "RNA flow cytometry". There are only three references[52-54] associated with RNA flow cytometry. More should be presented in section 5 or a separate section on the current state of RNA flow cytometry.

We thank you the reviewer for his/her comments. As requested also by other reviewers, we have added an Introduction paragraph explaining the structure of the review. Moreover, we have added a new Table 1 that includes several RNA Flow studies, while specific examples are mentioned in the text (lines 360-362 and 374-378).

Reviewer 4 Report

The paper consists of an interesting review that takes stock of T cell metabolism and how to study it on a "per cell basis", with particular regard to cytometric techniques. The biochemical approach is original, and the work is exhaustive and well written, but it seems a little unbalanced, inasmuch as, being a review, the cytometric techniques could be treated more exhaustively. The protocol presented is interesting, but perhaps the implications of the results could be discussed in more detail, and data analysis techniques, which are only hinted at, could also be more deeply discussed. There are other little things, including Figure 1, which is too schematic and not very informative, and a truncated sentence on line 177.

Author Response

The paper consists of an interesting review that takes stock of T cell metabolism and how to study it on a "per cell basis", with particular regard to cytometric techniques. The biochemical approach is original, and the work is exhaustive and well written, but it seems a little unbalanced, inasmuch as, being a review, the cytometric techniques could be treated more exhaustively.

We thank the reviewer for his/her comments. We have added a new Table 1 that includes several RNA Flow studies, while specific applications are mentioned in the text (lines 360-362 and 374-378). Moreover, Figure 1 has been implemented to include SmartFlare technology.

The protocol presented is interesting, but perhaps the implications of the results could be discussed in more detail, and data analysis techniques, which are only hinted at, could also be more deeply discussed.

We decided to show our data not with the aim to draw conclusions on Treg biology but only to display some representative results demonstrating feasibility and illustrating potential applications of the technique. However, we have added some comments regarding the interpretation of the data in paragraph 7 (lines 471-479 and 499-505).

There are other little things, including Figure 1, which is too schematic and not very informative,

Figure 1 has been implemented to include SmartFlare technology.

and a truncated sentence on line 177.

The truncated sentence has been corrected.